# On the Impact of Hyper-Parameters on the Inference Time Privacy of Deep Neural Networks

## Abstract

The deployment of deep neural networks (DNNs) in many real-world applications leads to the processing of huge amounts of potentially sensitive data. This raises important new concerns, in particular with regards to the privacy of individuals whose data is used by these DNNs. In this work, we focus on DNNs trained to identify biometric markers from images, e.g., gender classification, which have been shown to leak unrelated private attributes at inference time, e.g., ethnicity, also referred to as unintentional feature leakage. Existing literature has tackled this problem through architecture specific and complex techniques that are hard to put into place in practice. In contrast we focus on a very generalizable aspect of DNNs, the hyper-parameters used to train them, and study how they impact the privacy risk. Specifically, we follow a multi-fidelity and multi-objective HPO approach to (i) conduct the first empirical study of the impact of hyper-parameters on the risk of unintended feature leakage (inference time privacy risk); (ii) demonstrate that, for a specific main task, HPO successfully identifies hyper-parameter configurations that considerably reduce the privacy risk at a very low impact on utility, only by changing hyper-parameters; and (iii) evidence that there exist hyper-parameter configurations that have a significant impact on the privacy risk, regardless of the choice of main and private tasks, i.e., hyper-parameters that generally better preserve privacy blueunder our threat model and within the evaluated biometric setting.

## 1 Introduction

Deep Neural Networks (DNNs) have led to tremendous strides in visual recognition such image classification He et al. (2016), object detection Ren et al. (2015), semantic segmentation Chen et al. (2018), human pose estimation Kanazawa et al. (2018), and 3D reconstruction Dou et al. (2017). As such, they have now reached the point of being deployed in many real-world applications, such as automated driving, surveillance and security, and image content analysis in social media.

Many models performing such tasks have been deployed on the cloud and monetized as Machine Learning as a Service (MLaaS) for which users send their data to a server and get a prediction as a result. These services, however, incur the need for users to send their private data to an untrusted server, through a potentially untrusted network. User data, and in particular images, contain a lot of private information that is not necessarily related to the main task offered as a service. Model partitioning (Osia et al., 2018; Chi et al., 2018; Osia et al., 2017), separating the model into an embedding network and an inference head, has therefore emerged as a solution to mitigate these privacy concerns by locally computing embeddings representing their private data using the embedding network. These embeddings can then be sent to the server, which performs inference on them, making it harder for the server or a potential adversary to infer information unrelated to the main task. While not sending private data directly to untrusted third-parties, these locally-computed embeddings are still very vulnerable to unintended feature leakage (Melis et al., 2018; Özbulak et al., 2016; Das et al., 2018; Parde et al., 2019; Terhörst et al., 2020; Song & Shmatikov, 2020). That is, a passive adversary having access to the computed embeddings may still be able to infer auxiliary information unrelated to the main task. This shows that model partitioning and relying on embeddings that

represent the original user data is not enough to prevent inference of attributes that the user may not have consented to share, as depicted in Fig. 1.

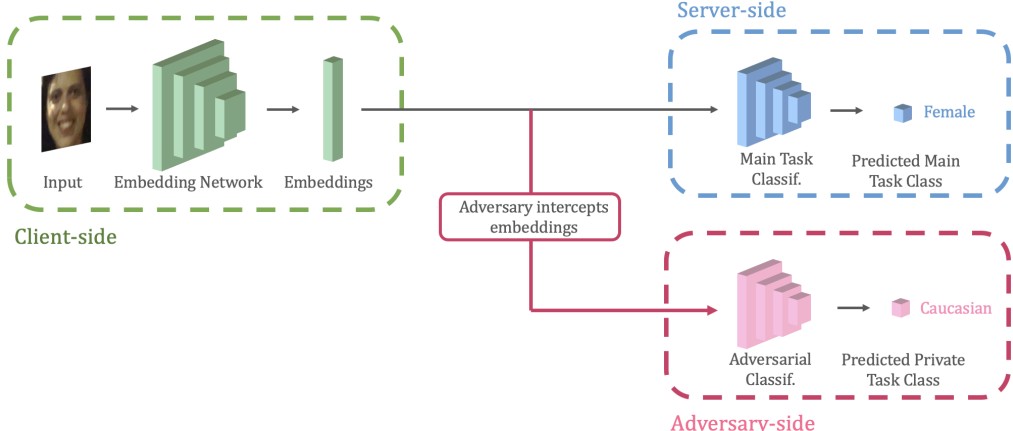

Figure 1: Example of unintended feature leakage in a cloud service for gender classification. Under our described adversarial model, the adversary is able to predict the private attribute "Ethnicity" when accessing to the embeddings output by the Embedding Network on the client side.

In this paper, we therefore place ourselves in the adversarial scenario where an attacker is able to intercept the embeddings extracted by a DNN trained for a specific task. This adversarial scenario, which focuses on the privacy leakage of embeddings at inference time, is common in the privacy-preserving representation learning field (Li et al., 2019; Stadler et al., 2024; Osia et al., 2017; 2018). Previous works have tackled unintended feature leakage in this context via adversarial and disentangled representation learning (Bertran et al., 2019; Roy & Boddeti, 2019; Li et al., 2021; Wang et al., 2024; Morales et al., 2021; Bortolato et al., 2020). However, the resulting methods are usually highly task-specific and require a specific architecture or training process, which complicates their generalization to other tasks. Indeed, deploying these techniques in cloud services will not be seamless, deterring service providers from putting them into practice, especially if privacy is not a priority. Additionally, all mitigation methods come with an inherent privacy-utility trade-off, where better privacy comes at the cost of main task accuracy, and which is very difficult to tune in practice.

By contrast, we focus on a highly-generalizable aspect of DNNs, which is architecture agnostic: The hyper-parameters used to train these DNNs. The impact of hyper-parameters on the risk of unintended feature leakage remains **largely unstudied**. We therefore ask ourselves the following questions: Can we identify hyper-parameters that have the most impact on privacy risk, and if so, do they generalize to other tasks or model architectures? This would enable the training of DNNs whose embeddings leak the least amount of private information possible while preserving what is necessary for the main task, therefore limiting the risk of unintended feature leakage.

Hyper-parameters have been shown countless times to have a significant impact on the accuracy of the main task (Wong et al., 2019; Bergstra et al., 2013; Hutter et al., 2013; 2014; Liao et al., 2022), which has led to the development and surge of hyper-parameter-optimization (HPO) methods (O'Malley et al., 2019; Falkner et al., 2018), aiming to find the hyper-parameter configurations incurring the best performing models in terms of accuracy. Additionally, hyper-parameters have also been shown to have an impact on other metrics such as fairness or energy consumption (Sukthanker et al., 2023). In this paper, we study the impact of hyper-parameters on **privacy** and leverage multi-objective HPO (Knowles, 2006) to find Pareto-dominant models in terms of both main task accuracy and privacy.

We therefore hope to guide the ML community and industry towards more privacy-preserving models with the following **contributions**:

1. Focusing on DNNs whose main task is a biometric classification task, e.g., gender classification, we conduct the **first empirical study** of the impact of hyper-parameters on the risk of unintended feature leakage at inference time(privacy risk).

2. We introduce the first use of a multi-objective HPO strategy to both maximize the main task accuracy and minimize the risk of unintended feature leakage. Introducing a more complex privacy metric to existing meta-learning techniques, we show that it is possible to train models that better preserve privacy at a negligible utility cost. Using this method, we are able to **discover general rules about hyper-parameter configurations** that better preserve privacy, and enable the training of embeddings networks **without relying in costly methods**.

3. We observe that there exist hyper-parameter configurations that have a significant impact on the privacy risk, regardless of the choice of main and private tasks, i.e., hyper-parameters that generally better preserve privacy.

We will make our code publicly available.

## 2 Related Work

### 2.1 Privacy-Preserving Machine Learning

Privacy applied to machine learning (ML) covers a broad range of adversarial models, privacy risks and mitigation techniques. The most commonly explored aspect in the field are those that pertain to the training set privacy, as training samples can be reconstructed from gradients (Guo et al., 2025), samples' membership can be inferred (Hu et al., 2022), or attributes can be recovered Gong & Liu (2018), among other. In contrast, inference time privacy, which is our focus in this paper, is considerably less explored. This is especially relevant in the context of our collaborative inference set-up, where a user sharing features with a server during inference, could lead to many privacy risks, such as unintended feature leakage (Melis et al., 2018; Özbulak et al., 2016; Das et al., 2018; Parde et al., 2019; Terhörst et al., 2020; Song & Shmatikov, 2020) or inference time model inversion attacks (He et al., 2019; 2020; Yang et al., 2022).

Previous works have tackled unintended feature leakage in collaborative inference by developing mitigation methods such as those that follow an adversarial (Bertran et al., 2019; Roy & Boddeti, 2019; Li et al., 2021; Morales et al., 2021) or disentangled (Bortolato et al., 2020; Wang et al., 2024) representation learning strategy, aiming to retain information about the task of interest while discarding that about the private soft-biometrics; as well as methods that focus on obfuscating specific features that carry private information while not being useful for the main task(Mireshghallah et al., 2019; 2021; Singh et al., 2021). While effective, these techniques suffer from drawbacks, such the fact that they suffer from a trade-off between main task accuracy and privacy, as most of these methods degrade utility significantly. More importantly, most of the existing methods are very task specific or hard to deploy in practice, for example because they use embedding networks with very specific architectures (Bortolato et al., 2020; Wang et al., 2024), or complex adversarial training methods that rely on hard-to-tune hyper-parameters (Sukthanker et al., 2023). By contrast, we focus on identifying the configurations of hyper-parameters that minimize the privacy risk, and without modifying the desired architecture, or change the training strategy beyond hyper-parameters. While finding the optimal hyper-parameters for a given architecture through HPO is **costly**, our large-scale empirical study on the impact of hyper-parameters on privacy allows us to define **general rules on the most privacy-preserving configurations, thus considerably facilitating the deployment process.**

Some previous works have studied the impact of hyper-parameters in the context of privacy, but focus on very different privacy tasks. For example, Arous et al. (2023) and Tan et al. (2022) study the impact of hyper-parameters and over-parameterization on membership inference attacks (MIA), which aim to infer if a target data sample was used to train the model or not. Hyper-parameters have been shown to be directly linked to overfitting, which in turn has been proven to directly impact the success of MIA attacks and therefore the privacy of members of the training set (van Breugel et al., 2023). Although motivated by these results, we explore the impact of hyper-parameters on entirely different types of privacy risks, namely **unintended feature leakage**. This is a very different setup, as we do not focus on privacy w.r.t. the

training set, but on the privacy risk at **inference time**. In contrast with the rather intuitive results for MIA, the link between training hyper-parameters and the risk of unintended feature leakage at inference time remains **largely unexplored.**

Developing privacy-preserving technology comes with an inherent privacy-utility trade-off, which is impossible to avoid. This is demonstrated in (Stadler et al., 2024), which claims that "it is not possible to learn representations that have high utility for the intended task but, at the same time, prevent inference of any attribute other than the task label itself". Regardless, we propose a method that identifies the training hyper-parameters of DNNs that will decrease the privacy risk *as much as possible* while maintaining a very similar main-task accuracy.

Additionally, we make the privacy-utility trade-off observable, therefore enabling more educated decisions about how much privacy is considered when developing models.

### 2.2 Hyper-parameter Optimization (HPO)

HPO refers to the field aiming to automate the search for the optimal hyper-parameters, including batch size, learning rate, optimizer, loss function, and in some cases architectural choices. Work on HPO has mostly focused on finding training strategies to maximize performance (Liu et al., 2019; Nekrasov et al., 2019). Some methods have nonetheless been developed for multi-objective HPO, to also optimize for a secondary objective, such as power consumption, model size, latency, or even fairness (Tian et al., 2021). In particular, (Sukthanker et al., 2023) jointly uses HPO and neural architecture search to optimize for both performance and fairness. However, to the best of our knowledge, such studies have never been conducted with privacy as a goal.

## 3 Method

### 3.1 Multi-Objective and Multi-Fidelity HPO

Our goal is to find hyper parameters with the biggest impact on privacy risk and therefore enable us to train models that minimize the risk of unintended feature leakage the most. To train a sufficent amount of models and explore the hyper-parameter space efficiently, we leverage HPO techniques to find hyper-parameter configurations that maximize utility and minimize privacy risk. This will provide us with a diverse set of models trained with many combinations of hyper-parameters which will then allow us to conduct a large scale empirical study of the impact of hyper-parameters on privacy risk.

Traditional HPO focuses on finding the hyper-parameter configurations that lead to the models with the best main task accuracy. Here, however, we seek to optimize for *both* accuracy and privacy. Doing so requires the following two properties: **(i) Multi-objective**: in contrast to classical HPO that optimizes for main task performance only, the goal is to optimize for two or more objectives, this would enable the observation of the trade-off between accuracy and privacy; **(ii) multi-fidelity**: evaluating the accuracy and privacy of a hyper-parameter configuration requires fully training a deep neural network. This can be computationally expensive, which motivates the need to approximate the conventional evaluation methods.

In our study, we use the SMAC3 package (Lindauer et al., 2021) that offers a framework satisfying both properties using a HyperBand-based algorithm (Li et al., 2016) called Sequential Surrogate Model-Based Optimization (Audet et al., 2000) for multi-fidelity and using the ParEGO algorithm (Knowles, 2006) for multi-objective. We describe in more detail in Appendix A.1.1 the algorithms used by the SMAC package for the multi-objective and multi-fidelity optimization.

### 3.2 Metrics

Before exploring the impact of hyper-parameters on the utility-privacy trade-off of a model for a specific pair of main and private tasks, we need to clearly define the aforementioned costs we will be using throughout this paper and which will be introduced into multi-objective HPO. As depicted in Fig 1, a model trained for a biometric task is usually divided into two parts. First, an embedding network, taking as input an image of

a face and returning an *embedding* in a latent space. Second, a head, also called classifier, which is a small network that, given an embedding, returns probabilities that the sample belongs to a given set of classes. Embeddings represent the information extracted by the model before it uses the head to make a prediction on the main biometric task. We study specifically the privacy leakage of the embeddings, the output of the first part of our network, and how much information they leak about a chosen private task.

To measure the utility of such a model, we simply use the **main task accuracy** as our metric. Specifically, for a given biometric task, we measure the proportion of the test set samples that are correctly classified by the model. Ideally, we would like to *maximize* utility, and therefore maximize the accuracy on the main task.

Measuring privacy, however, is more involved, and how to accurately measure the privacy risk of a given system is acknowledged as a one of the core, and arguably most complex, aspects of privacy research. It requires modeling an adequate and realistic adversary by defining what information and resources it has access to, i.e, its capabilities. This is complicated by the difficulty to prove that the chosen adversarial model represents the best possible adversary, and mis-identifying its capabilities could lead to vulnerabilities. Furthermore, even with a good adversary, one needs to define the properties that are measured by the privacy metric, such as uncertainty, information gain or loss, or probability of the adversary's success, among others (Wagner & Eckhoff, 2018).

Following standard practice, we draw inspiration from the Bayesian-optimal adversary literature (Sablayrolles et al., 2019; Stadler et al., 2024; Chatzikokolakis et al., 2020) to measure privacy. A Bayesian-optimal adversary is the adversary that has the best accuracy on the private task. Ideally, we would like to measure the risk of unintended feature leakage by measuring the probability of success of this optimal adversary $\mathcal{A}(X)$ trying to predict sensitive attribute $Z$ from the set of sensitive classes $\mathbb{Z}$ when given access to embedding $X$ in latent space $\mathbb{X}$. In this formalism, the worst-case risk of unintended feature leakage is thus expressed as

$$\hat{Z}(X) = \underset{\mathcal{A}:\mathbb{X}\to\mathbb{Z}}{\operatorname{argmax}} \mathbb{P}[Z = \mathcal{A}(X)]. \tag{1}$$

This optimal adversary can then be used to measure the privacy risk (or privacy loss) of sensitive attribute $Z$ as

$$PR = \mathbb{P}[Z = \hat{Z}(X)] - \mathbb{P}[Z = \hat{Z}], \tag{2}$$

where $\hat{Z}$ without any argument represents the baseline guess of optimal adversary $\mathcal{A}$ on sensitive attribute $Z$ when it does not have access to any embedding. The privacy risk therefore approaches zero when the optimal adversary's prediction is close to the random guess, which means that observing embedding $X$ reveals virtually no information about sensitive attribute $Z$. Ideally, we would therefore like to *minimize* the privacy risk $PR$, which is equivalent to minimizing $\mathbb{P}[Z = \hat{Z}(X)]$. In other words, we seek to measure the privacy risk as the **accuracy of an optimal adversary trying to predict the private attribute** given an embedding.

In practice, finding the best possible adversary by solving problem 1 to optimality cannot be guaranteed, and we therefore aim to find the best possible *approximation* of the optimal adversary $\mathcal{A}$. We achieve this by making the assumption that our adversary has **query access** to the embedding network, i.e., it can query it as many times as desired with face images and obtain the resulting embeddings, without having a white-box view of the model. Additionally, we assume that the adversary has access to a dataset with a **similar distribution** to the potential queries of the model, i.e., face images, labeled with various attributes of interest (ethnicity, gender, age, etc.), and uses it in conjunction with the embedding network to learn how to predict private information from face embeddings. Then, given a model trained for a main task, we use hyper-parameter optimization to find the hyper-parameters of the adversarial model that yield a strong enough adversary. We will show empirically that the precise choice of the adversary in this set-up has limited impact on our method, as long as it is sufficiently strong.

## 4 Experiments

### 4.1 Experimental Set-Up

Table 1: Experimental Setup and Search Space

| | Experimental Setup |
|---:|:---|
| **Datasets** | FairFace (Kärkkäinen & Joo, 2019), |
| **Embed. Net Architecture** | VGG16 (Simonyan & Zisserman, 2014) |
| **Adversarial Classifier** | FC layers of VGG16, Linear, CosFace (Wang et al., 2018), ArcFace (Deng et al., 2019) |
| **HPO Metrics** | Main Task Accuracy↑ & Private Task Leakage Risk ↓ |
| **# HPO trials** | 80 |
| **Min and Max Budgets (Epochs)** | $[5, 120]$ epochs |
| **Drop Ratio** | 3 |
| | **Search Space** |
| **Batch Size** | $[30, 100]$ with a step of size 5 |
| **Head** | $\{\text{Linear}, \text{CosFace (Wang et al., 2018)}, \text{ArcFace (Deng et al., 2019)}\}$ |
| **Loss** | $\{\text{Cross-Entropy (Mao et al., 2023)}, \text{Focal Loss (Lin et al., 2018)}\}$ |
| **Optimizer** | $\{\text{Adam (Kingma & Ba, 2017)}, \text{SGD (Amari, 1993)}\}$ |
| **Learning Rate** | $[10^{-5}, 10^{-1}]$ if Adam (log scale) 
 $[10^{-4}, 1]$ if SGD (log scale) |
| **Weight Decay** | $[10^{-6}, 10^{-1}]$ (log scale) |

We use the SMAC3 package (Lindauer et al., 2021) offering the optimization framework described in the previous section. Below, we define the DNN architectures we focus on, as well as the search space and the costs we seek to optimize for, as summarized in Table 1. The SMAC3 configuration we used is detailed in Appendix A.1.2.

#### 4.1.1 Architecture and Dataset

Following previous work (Li et al., 2019; Singh & Shukla, 2021; Abbasi et al., 2024), we use a CNN architecture, VGG16 (Simonyan & Zisserman, 2014), for the embedding network, with a latent space of size 4608, excluding the final fully-connected layers of the architectures, which we employ as our adversarial classifier and as one of the options for the main task classifier, the other options being CosFace (Wang et al., 2018) and ArcFace (Deng et al., 2019). We additionally include the following hyper-parameters to our HPO search space: Batch size, loss, optimizer, learning rate, and weight decay.

We evaluate our approach and the baselines on the FairFace dataset (Kärkkäinen & Joo, 2019), which contains 108,501 face images balanced in gender, ethnicity and age. The main and private tasks are therefore chosen from the following soft-biometrics labels: Gender, Ethnicity, and Age, which have 2, 7, and 9 classes, respectively. We run our HPO set-up on 6 different combinations of main and private tasks, using the VGG architecture. In Appendix A.2, we report additional experiments on the more recent Inception-ResNet architecture (Szegedy et al., 2016) and the CelebA dataset (Liu et al., 2015).

#### 4.1.2 Adversarial Model Construction

Before running an HPO for a specific pair of main and private task, we first search for the best possible adversary to measure the privacy risk as accurately as possible, as described in Section 3.2. To do so, we first train the embedding network for its main task for half of the maximum number of epochs, using a random configuration of hyper-parameters. This will result in an embedding network that is most likely not as accurate on the main task as a model fully trained using an optimal hyper-parameter configuration. As such, we expect the private task to be harder to infer, as less utility leads to more privacy (Zhong & Bu, 2022; Stadler et al., 2024). Additionally, Fig. 2a shows that training an adversary and measuring the privacy risk on the same embedding network at different training epochs (13, 40 and 120 epochs) yields a similar privacy risk. We observe similar results for other combinations of main and private tasks and architectures. This implies that an embedding network learns information about the private task early in its training, and we can therefore find a sufficiently strong adversary on a embedding network trained for half of the maximum

number of epochs. We then freeze this embedding network and run HPO on the adversarial classifier to find the hyper-parameters that maximize the adversary's accuracy on the private task. The search space for the adversary includes learning rate, batch size, loss function, and head. We limit the choice of the head architecture of the adversary to the same architecture as the head of the main task, as well as similar architectures (ArcFace, CosFace, Linear). (Li et al., 2019) shows empirically that, in practice, choosing the same architecture for the adversary classifier as the main task classifier (grey-box access) is a good strategy, and leads to a strong adversary. We assume that the adversary has a uniform prior, and that it has access to a dataset of similar distribution as the inference samples when training its attack model.

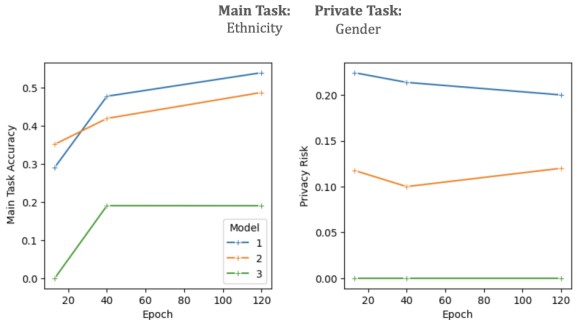 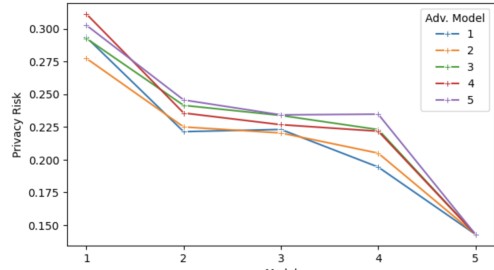

(a) Ablation study of the main task accuracy and privacy risk of 3 embedding networks measured at 3 different training epochs (13, 40, 120) by an adversary trained with the same HP configuration.

(b) Ablation study of the privacy risk of 5 embeddings networks as measured by 5 adversarial models trained with randomly sampled HP configurations.

Figure 2: Ablation study validating the adversary construction to measure privacy on a VGG-16 architecture trained with Gender as the main task and Ethnicity as the private task.

We acknowledge that this strategy to build an adversary that is used to measure privacy risk does not guarantee finding the best possible adversary. However, it yields a good proxy from which we can draw valuable conclusions. We validate this in Fig. 2b, which shows that for 5 models with the same architecture trained on the same main task but with different hyper-parameters, different adversaries with different architectures and trained with different hyper-parameters (1) always output, for the same embedding network, a similar privacy risk with a small standard deviation; (2) will always rank the embedding networks by privacy risk in a very similar manner, preserving the relative ordering of the models. These two properties are crucial for our multi-HPO approach, as they imply that we do not need guarantees of having the best possible adversary to chose the models that offer a better privacy than the other models trained.

We then use these hyper-parameters to re-train from scratch the adversarial classifier every time we measure the privacy risk on the private task during the main HPO run. Finally, once we have found the best hyper-parameter configuration for our embedding network through our multi-HPO approach, we re-measure privacy one last time by finding again the best adversary through HPO for our final embedding network. For each result, we report the mean and standard error over 3 seeds.

## 4.2   Baselines

We compare the model trained with the best configurations of hyper-parameters against the following baselines:

**Single-Objective HPO**: Our first baseline consists of simply running HPO using the same framework and set-up as for our approach but by optimizing for the **main task accuracy only**, as in traditional HPO. This optimization thus ignores privacy.

**Random Gaussian Noise**: For our second baseline, following common practice, particularly in Federated Learning (Liu et al., 2020; Papernot et al., 2018; Truex et al., 2018), we perturb the embeddings using random Gaussian noise, tuning $\sigma^2$ sufficiently to bring the privacy risk close to our method.

**Principal Component Analysis and Siamese Fine-tuning**: For our third baseline, we use the method described in (Osia et al., 2020), which uses a combination of fine-tuning, principal component analysis (PCA) dimensionality reduction and Laplace noise injection. This method first performs siamese fine-tuning (Chopra et al., 2005) on a model pre-trained for the main task. PCA dimensionality reduction is then applied to the resulting embeddings, before applying noise drawn from a Laplace distribution with scale $b = 2$, and sending these compressed and noisy embeddings to the server. Before inference, the server will decompress the noisy embeddings and es it through the main task classifier. We evaluate our method against two versions of this baseline, one with siamese fine-tuning and one without.

**Disentangled Representation Learning**: For our fourth baseline, we employ the Disentangled Representation Learning (DRL) algorithm presented in (Bortolato et al., 2020). This method consists of training a small encoder model (trained in an encoder-decoder fashion) that computes features that disentangle the main task from the private task.

**Cloak**: The fifth baseline we use is a method called **Cloak** presented in (Mireshghallah et al., 2021). This gradient based perturbation method learns a subset of features that carry less information about the main task and suppresses it.

**Adversarial Representation Learning**: Finally, we evaluate our method against the Adversarial Representation Learning (ARL) algorithm presented in (Li et al., 2019). The proposed learning algorithm is designed as a game between an embedding module, a main task classifier and a discriminator (which acts as a proxy adversary), where the embedding module tries to maximize the main task classifier's accuracy while minimizing a discriminator's ability to infer private attributes from the embeddings.

For every baseline, as for our method, we measure privacy by finding the best adversary through HPO for each embedding network. This lets us compare the effect of hyper-parameters on privacy risk to the impact of other methods specifically tailored to reduce the privacy risk.

## 4.3 Results & Discussion

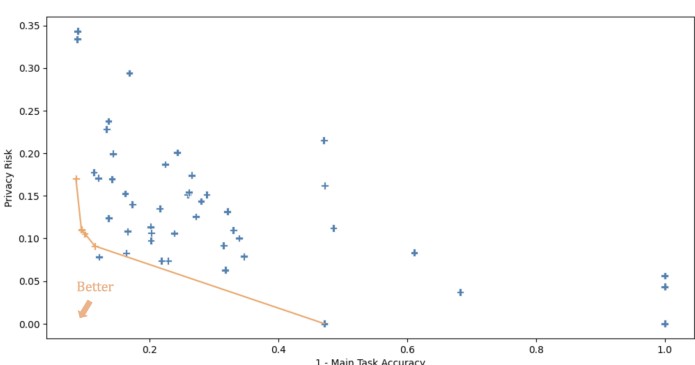

Figure 3: Costs and Pareto-front of the configurations that were trained during one instance of multi-objective hyper-parameter optimization, using the VGG-16 architecture, with gender classification as main task and ethnicity classification as private task.

As an example of the typical output of a multi-objective HPO run, in Fig. 3, we plot the costs of the configurations that were trained during an HPO run with the main task of gender recognition and private task of ethnicity, i.e., all configurations that were trained during their successive-halving bracket. The orange points connected by a line represent the Pareto-front of this optimization run: All the points whose costs are Pareto-dominant. A configuration is Pareto-dominant if no objective can be improved without sacrificing at least one other objective. Only configurations that were trained up to the maximal budget are considered as part of the Pareto-Front. Plotting the Pareto-front makes it possible to manually make decisions on what hyper-parameter configuration to chose among the Pareto-dominant ones: The privacy-accuracy trade-off decision is therefore made on a much smaller set. Ideally we would want to reach a point situated in the bottom left of the plot: High main task accuracy and low privacy risk. Note that for the same small range of main task accuracy (between 0.1 and 0.2 on the plot), we have a wide range of privacy risks. This confirms that f**or the same main task accuracy, there exist models that offer close to no privacy for a given private attribute, as well as models that yield a much more acceptable privacy risk.** By simply varying some hyper-parameters during training, we were able to find models with almost the same main

Table 2: Privacy-Utility trade-off comparison on the FairFace dataset using VGG16. Higher MA (↑) is better and lower PR (↓) is better.

| Method | Main Task → Private Task | | | | | | | | | | | |
|---|---|---|---|---|---|---|---|---|---|---|---|---|
| | Gender → Ethnicity | | Gender → Age | | Ethnicity → Gender | | Ethnicity → Age | | Age → Gender | | Age → Ethnicity | |
| | MA ↑ | PR ↓ | MA ↑ | PR ↓ | MA ↑ | PR ↓ | MA ↑ | PR ↓ | MA ↑ | PR ↓ | MA ↑ | PR ↓ |
| HPO | **0.917**±0.0010 | 0.323±0.0028 | **0.916**±0.0007 | 0.230±0.0047 | **0.667**±0.0012 | 0.173±0.0049 | **0.671**±0.0021 | 0.155±0.0065 | **0.512**±0.0019 | 0.212±0.0075 | 0.508±0.0001 | 0.237±0.0033 |
| Noisy | 0.889±0.0005 | 0.084±0.0014 | 0.889±0.0016 | 0.134±0.0008 | 0.661±0.0005 | 0.038±0.0008 | 0.663±0.0009 | 0.037±0.0009 | 0.410±0.0012 | 0.071±0.0044 | 0.404±0.0012 | 0.044±0.0017 |
| DL | 0.914±0.0006 | 0.257±0.0014 | 0.914±0.0003 | 0.258±0.0068 | 0.658±0.0005 | 0.157±0.0034 | 0.657±0.0004 | 0.149±0.0063 | 0.482±0.0082 | 0.236±0.0054 | 0.445±0.0153 | 0.198±0.0055 |
| ARL | 0.873±0.0282 | 0.316±0.0443 | 0.660±0.0267 | 0.253±0.0160 | 0.640±0.0045 | 0.248±0.0044 | 0.525±0.0320 | 0.314±0.0092 | 0.470±0.0045 | 0.261±0.0045 | 0.368±0.0049 | 0.382±0.0019 |
| PCA | 0.895±0.0037 | 0.085±0.0026 | 0.895±0.0027 | 0.137±0.0065 | 0.663±0.0001 | 0.041±0.0004 | 0.661±0.0011 | 0.043±0.0007 | 0.419±0.0010 | 0.073±0.0046 | 0.422±0.0028 | 0.051±0.0032 |
| S-PCA | 0.679±0.0194 | **0.020**±0.0055 | 0.684±0.0153 | **0.037**±0.0077 | 0.108±0.0458 | **0.003**±0.0025 | 0.159±0.0033 | 0.003±0.0011 | 0.372±0.0112 | **0.054**±0.0016 | 0.369±0.0122 | 0.046±0.0106 |
| Cloak | 0.604±0.0016 | 0.095±0.0021 | 0.605±0.0047 | 0.069±0.0063 | 0.238±0.0099 | 0.073±0.0127 | 0.236±0.0084 | 0.065±0.0026 | 0.191±0.0011 | 0.097±0.0025 | 0.188±0.0049 | 0.093±0.0041 |
| **MO-HPO (Ours)** | 0.883±0.0007 | 0.096±0.0017 | 0.894±0.0012 | 0.120±0.0056 | 0.511±0.0054 | 0.133±0.0060 | 0.607±0.0016 | 0.165±0.0133 | 0.446±0.0083 | 0.177±0.0060 | **0.517**±0.0072 | 0.153±0.0102 |
| **MO-HPO+PCA** | 0.866±0.0011 | **0.020**±0.0007 | 0.851±0.0005 | 0.044±0.0009 | 0.463±0.0013 | 0.036±0.0011 | 0.280±0.0010 | **0.002**±0.0014 | 0.435±0.0016 | 0.078±0.0025 | 0.216±0.0012 | **0.002**±0.0016 |

task accuracy while offering much more privacy. This suggests that, since different hyper-parameters lead to different local minima, which have been shown to benefit from different properties, such as generalization (Keskar et al., 2017), these local minima may also display different privacy behaviors.

In Table 2, we detail the main task accuracy and privacy risk, computed as described in Section 3.2, for our method and the baselines, for the 6 combinations of main and private tasks described in Section 4.1.

These results first show that, when comparing our multi-objective HPO method to embedding networks that were trained solely to maximize main task accuracy (**HPO**), our approach decreases the privacy risk significantly (by 5 to 20 points) for all but one combinations of main and private tasks. Specifically, our method brings the accuracy of the adversarial classifier significantly closer to random guess, while only suffering from a decrease in main task accuracy of less than 5 points in almost all cases. In other words, our approach maintains a good utility for a much lower privacy risk.

In comparison to the Disentangled Learning baseline (**DL**), our approach yields slightly worse utility but at the gain of better privacy. In fact, the privacy of DL is not much better than that of the HPO baseline, although the latter ignores the private task.

When comparing our method to the Adversarial Representation Learning (**ARL**) algorithm, we observe that not only does the main task accuracy undergoes a larger drop than with our method, but the privacy risk is sometimes even slightly higher than that of models trained with no privacy taken into account. Indeed, although the accuracy of the adversarial classifier on the private task during training decreases significantly, this privacy gain does not transfer to using a freshly trained adversarial classifier to measure privacy. Our method, on the other hand, still maintains the same privacy gain when measuring privacy with freshly trained adversary. Additionally, these results highlight the complexity of choosing appropriate hyper-parameters for ARL algorithms, especially those managing the adversarial loss and the weighs of the different objectives.

We then compare our method to the PCA-Laplace baseline (**PCA**) and Gaussian Noise baseline (**Noisy**), which similarly offer the best privacy-utility trade-off out of our baselines. In most cases, this method provides slightly better privacy than our method (at most five points), as well as a slightly better accuracy, but by still offering a similar privacy-utility trade-off. This is a valuable insight, as it shows that **by only tuning hyper-parameters, one can reach similar privacy to a state-of-the-art method.** By contrast, the Siamese Fine-tuning variation of PCA-Laplace (**S-PCA**) offers very good privacy but at the cost of a significant drop in main task accuracy, making it a suboptimal choice of privacy risk mitigation technique.

The **Cloak** method leads to similar conclusions, where we reach a slightly smaller Privacy Risk that MO-HPO, but at a much bigger cost on utility.

Finally, we evaluate a combination of our method and the PCA-Laplace method by using a model trained using the hyper-parameter configuration discovered using our Multi-Objective HPO and applying to its embeddings PCA compression, Laplace noise injection (with a identical scale $b = 2$ for comparison purposes), and PCA decompression. This strategy (**MO-HPO-PCA**) yields a lower privacy risk than both MO-HPO and PCA taken individually, however in some cases has a significant hit on utility. This method would require reduce the scale in order to not degrade utility as much.

We report main task accuracy and privacy risk for our method and for the baselines on additional datasets and architectures in Appendix A.2.

Additionally, we investigate whether models trained using different privacy-preserving methods are protected against the inference of unrelated private attributes. For example, does an embedding network trained to maximize its accuracy on gender classification while minimizing the privacy risk with regards to ethnicity also protect against the inference of age? To study this, we use all the embedding networks trained in the previous set of experiments to minimize the privacy risk of a given private attribute and train an adversarial classifier to predict **another unrelated private attribute**. This lets us measure the privacy risk with respect to this unrelated attribute. We expose our findings in Table 3. When using our method, in most cases the privacy risk for the unrelated task is decreased significantly when compared to a classic embedding network trained with no regards to privacy. This evidences that training a model while trying to decrease the privacy risk with regards to a specific private attribute has a positive impact on the privacy risk of unrelated private attributes. The transferability of the selected hyper-parameter configurations to other private tasks also suggests that **the same hyper-parameters have a consistent impact on the privacy risk.** We investigate this further in the next section.

Table 3: Leakage of a private attribute unrelated to the private attribute used to train the model. The lower the privacy risk, the better.

|  |  |  | HPO | Multi-Obj. HPO (Ours) |
| --- | --- | --- | --- | --- |
| Main Task | Private Task | Unrelated Private Task | Privacy Risk ↓ | Privacy Risk ↓ |
| Gender | Ethnicity | Age | 0.440 | **0.248** |
| Gender | Age | Ethnicity | 0.438 | **0.218** |
| Ethnicity | Gender | Age | 0.267 | **0.242** |
| Ethnicity | Age | Gender | **0.670** | 0.680 |
| Age | Gender | Ethnicity | 0.356 | **0.276** |
| Age | Ethnicity | Gender | 0.758 | **0.679** |

### 4.4 Empirical Study of the Impact of Hyper-Parameters on Privacy Risk

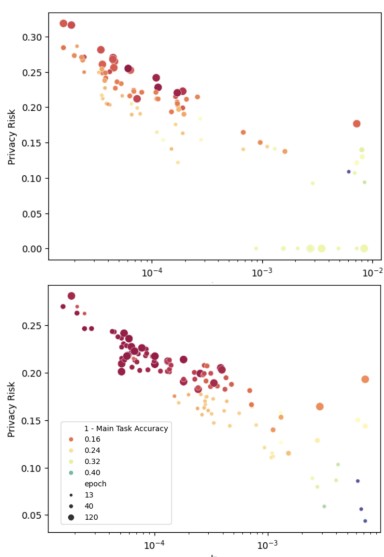

Figure 4: Privacy risk as a function of the learning rate for all models trained using the Adam optimizer during two runs of multi-objective HPO for two different combinations of main and private tasks: Age-Gender (top), and Ethnicity-Age (bottom).

The many multi-objective HPO runs we have performed across different combinations of main and private tasks have provided us with a large collection of embedding networks and their associated main task accuracy and privacy risk. We now leverage this data to investigate the impact of hyper-parameters on the risk of unintended feature leakage, **which has never been studied before to our best knowledge**.

To study the impact of hyper-parameters in a more isolated setting, we first consider the models that were trained in a slightly different multi-objective HPO setting. That is, instead of the 6 hyper-parameters described in the experimental set-up of Table 1, we reduce the search space to 4 hyper-parameters: Loss, learning rate, optimizer, and batch size. We therefore fix the weight decay parameter to $10^{-3}$ and the head to the FC Linear architecture. In Fig. 4, we plot the privacy risk as a function of the learning rate for all models trained using the Adam optimizer during two runs of multi-objective HPO for two different combinations of main and private tasks (Age-Gender at the top and Ethnicity-Age at the bottom). The size of the points indicates for how many epochs the model was trained, and the color depicts the performance on the main task (the closer to red, the better). These plots highlight a strong negative correlation between the learning rate and the privacy risk. Indeed, **the smaller the learning rate, the higher the privacy risk.** We confirm this by computing the Pearson correlation between the

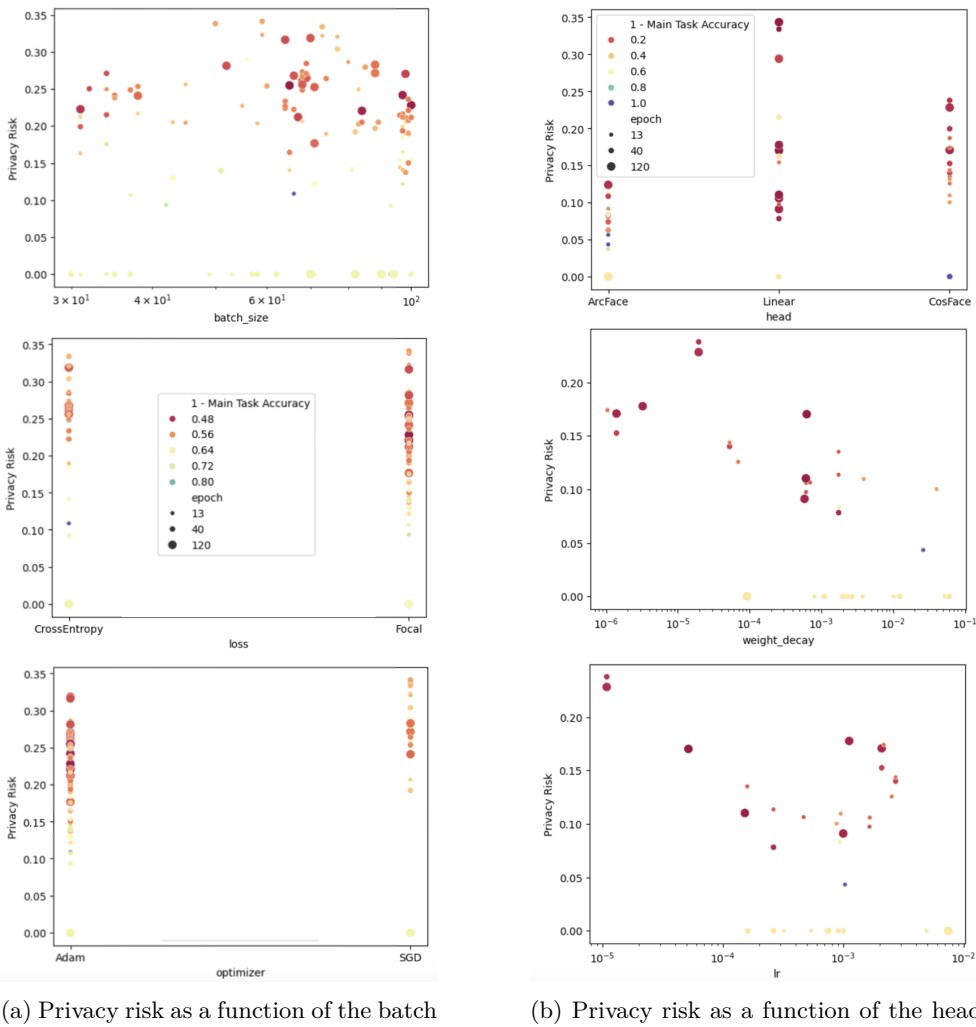

(a) Privacy risk as a function of the batch size, loss, and optimizer for all models trained during a run of multi-objective HPO on the VGG-16 model for the combination of main and private tasks Age-Gender. The size of the points indicates how many epochs the model was trained, and the color shows performance on the main task (the closer to red, the better).

(b) Privacy risk as a function of the head, weight decay, and learning rate for all models trained during a run of multi-objective HPO on the VGG-16 model for the combination of main and private tasks Age-Gender. Point size indicates training epochs, color shows main task performance.

Figure 5: Privacy risk under different hyperparameter combinations during multi-objective HPO on VGG-16 for the Age-Gender task combination.

learning rate and the privacy risk: -0.685 and -0.647 respectively for both plots. These plots further show that, as expected, the lower the privacy risk, the lower the main task accuracy in general. Nevertheless, some points with lower privacy risk still correspond to excellent main task accuracy; these are the hyper-parameter configurations discovered by our method. The observed trends for these two combinations of main and private task can also be observed across the other combinations we studied. We added an alternate version of Fig. 4 in A.3 for a clearer analysis.

The three other hyper-parameters, batch size, loss, and optimizer, seem to have no observable impact on the privacy risk. We illustrate this in Fig. 5a by plotting the privacy risk versus the batch size, loss, and optimizer for the models trained during a multi-objective HPO run with age as the main task and gender as the private task. These plots do not show any correlation between these three hyper-parameters and the privacy risk, which is confirmed by very small Pearson correlations (-0.056, -0.039, and 0.204) between the

privacy risk and each of the three hyper-parameters. The same trends are also observable for other main and private task combinations.

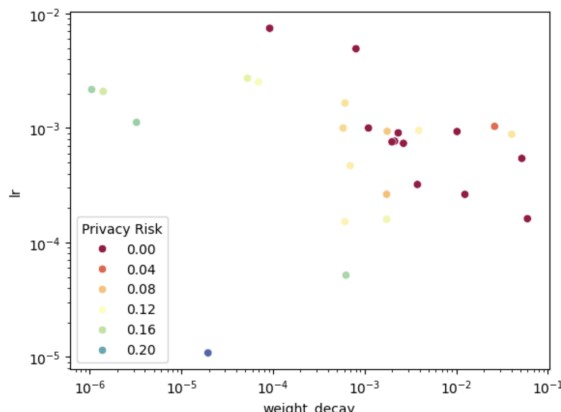

Figure 6: Privacy risk as a function of both learning rate and weight decay for all models trained during a run of multi-objective HPO on the VGG-16 model for the combination of main and private tasks Gender-Ethnicity. The color indicates the privacy risk (the closer to red, the better).

We now investigate the impact of the two hyper-parameters that were left out in the previous experiments, weight decay and head architecture, on the privacy risk and how they interact with the other hyper-parameters we already studied. That is, we go back to our initial search space of 6 hyper-parameters described in Table 1, and use the models from the 6 multi-objective HPO runs we conducted in the previous section. In Fig. 5b, we plot the privacy risk as a function of the head, weight decay, and learning rate for gender classification as the main task and ethnicity as the private attribute. The first plot in the figure evidences that the choice of the head has some impact on both privacy and main task accuracy. Indeed ArcFace seems to provide lower privacy risk than the other two head architectures, but at the cost of a lower main task accuracy. The Linear and CosFace head architectures provide similar main task accuracy, but the Linear head yields a larger range of privacy risks, ultimately resulting in models with a lower privacy risk than CosFace for a higher main task accuracy. The second plot shows a similar trend as for the learning rate, with a negative correlation between weight decay and privacy risk, i.e.,

a larger weight decay leads to a lower privacy risk. On the third plot, we can observe again that the learning rate has an impact on the privacy risk, although the trend is less evident than in Fig. 4, which showed a clear relationship between learning rate and privacy risk. This can be explained by the fact that the models depicted here vary in terms of both learning rate and weight decay, which makes it harder to study the impact of each individual hyper-parameter on the privacy risk.

Indeed, it is in fact **the combination of both that impacts the privacy risk the most**, as depicted in Fig. 6, which shows the privacy risk (represented by color) as a function of both the learning rate and privacy risk. This plot shows that a combination of a larger weight decay and learning rate leads to better privacy. Additionally, for a fixed weight decay of $10^{-3}$, varying the learning rate is clearly very negatively correlated with the privacy risk, which is exactly what we had observed in Fig. 4. We have also observed the same trends for the other combinations of main and private tasks.

## 5    Conclusion

In this paper, using multi-objective HPO, we have conducted the first empirical study of the impact of hyper-parameters on the risk of unintended feature leakage, and discovered that the learning rate in conjunction with the weight decay have the most impact on privacy. We have also shown how using hyper-parameters to find models that minimize privacy risk for a given private task with a limited impact of accuracy also transfers to other private tasks, evidencing further the generalization of our method. Overall, we have presented a new approach to training DNNs for biometric tasks in a privacy-preserving way while remaining easier to deploy in practice than previous, by relying on hyper-parameters as opposed to more complex methods. In future we aim to extend our empirical study to larger architectures such as foundation models (Caron et al., 2021; Oquab et al., 2024; Radford et al., 2021) as well as studying the impact of architecture on privacy risk using multi-objective Neural Architecture Search (NAS) instead of HPO, which seeks to automate the search for the best possible architecture for a task at hand (Wang, 2021; Liu et al., 2019; Nekrasov et al., 2019; Yu et al., 2020; 2021), and which we suspect will also have significant impact on privacy risk.

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

# A    Appendix

## A.1    SMAC3: Multi-Objective and Multi-Fidelity HPO

### A.1.1    Algorithms

Traditional HPO focuses on finding the hyper-parameter configurations that lead to the models with the best main task accuracy. Here, however, we seek to optimize for *both* accuracy and privacy. Doing so requires the following two properties:

- **Multi-objective**: In contrast to classical HPO that optimizes for main task performance only, the goal is to optimize for two or more objectives. This would enable the observation of the trade-off between accuracy and privacy.

- **Multi-fidelity**: Evaluating the accuracy and privacy of a hyper-parameter configuration requires fully training a deep neural network. This can be very computationally expensive, which motivates the need to approximate the conventional evaluation methods.

Similarly to Sukthanker et al. (2023), we use the SMAC3 package (Lindauer et al., 2021) that offers a framework satisfying both properties. For multi-fidelity, a HyperBand-based algorithm (Li et al., 2016) is used: It aims to make optimization more efficient by smartly allocating a budget (in this case the budget is epochs) to a set of hyper-parameter configurations, referred to as a *bracket*, evaluating the performance of all configurations, throwing out the worst half, and repeating until only a few configurations remain. This limits the number of models that are actually fully trained, significantly speeding up the optimization time. Multiple brackets are launched in parallel, each with a different number of starting epochs and ratio of models discarded at each step, improving the exploration-intensification trade-off.

The multi-objective aspect of the method then appears when deciding what next configuration to generate when launching a new bracket. Rather than randomly choosing the next configuration in the search space, the algorithm uses Sequential Surrogate Model-Based Optimization (Audet et al., 2000). In short, the optimizer keeps a surrogate model that, given a configuration, outputs how well it performs. This surrogate model is regularly retrained using the results of the previous brackets. The optimizer can then select new configurations in two ways: Either using the surrogate model, which will focus on configurations close in the search space to well-performing ones, *or* by randomly choosing a configuration from unexplored regions of the search space. This strategy yields a good trade-off between intensification and exploration.

The optimizer uses the ParEGO algorithm (Knowles, 2006) to train its surrogate model. This algorithm addresses the question of how to quantify a good performing configuration in the presence of two or more objectives. To do so, every time the surrogate model is retrained, a new cost function is generated as follows: Given a list of $n$ costs, $cost_j$ for $j \in [1, n]$ and parameter $\rho$ (default 0.05), the algorithm computes the cost function

$$\max_j \left[ cost_j \times \theta_j + \rho \sum_{i=1}^{n} cost_i \times \theta_i \right], \tag{3}$$

where the $\theta_i$s are uniformly sampled in $[0, 1]$ upon instantiation and re-sampled every time we retrain the surrogate model. This cost function ensures that the surrogate model gives more importance to a different objective randomly every time it is retrained, enabling exploration in different directions of the configuration space.

### A.1.2 SMAC3 Configuration

For each run, we aim to maximize the accuracy on the main task while minimizing the adversary's accuracy on the private task. We run HPO with a maximum budget of 120 epochs and for 80 trials (a trial being the training from scratch *or* the intensification of a model for which the training has already been done for a limited budget). Finally we choose a drop ratio of 3, meaning that, every 1/3 of the maximum number of epochs, we get rid of 1/3 of the remaining configurations in the bracket. The optimizer finally outputs a list of the Pareto-dominant configurations in terms of main task accuracy and privacy risk.

### A.2 Additional Experiments: Visual Examples and Additional Datasets and Architectures

We first showcase visual examples of how our method compares to two baselines in Fig. 7. The input images are passed to three embedding networks; one trained using our multi-objective HPO method, one using HPO to maximize the main task accuracy only, and the last one implementing the gaussian noise injection baseline. The embedding obtained with our approach still yields a correct main task prediction, even though the embedding network was trained in a privacy-preserving manner (the genders of the input images are still

predicted accurately). However, this embedding successfully fools a strong adversarial classifier, as opposed to that obtained with the baseline, which reveals the ethnicity of the input image, thus resulting in a higher privacy risk. On the other hand, the noise injection baseline also provides a very low privacy risk but suffers from a significant loss in utility (the genders of the input images are no longer successfully predicted in most cases). We can observe a similar privacy-utility trade-off for these three methods with another combination of private and main tasks, age and gender, depicted on the right of Fig. 7.

| Main Task Label | Female | Male | Male | | Main Task Label | 30-39 | 3-9 | 40-49 |
|---|---|---|---|---|---|---|---|---|
| Priv. Task Label | White | East-Asian | Black | | Priv. Task Label | Female | Female | Female |
| **HPO** - Good Utility Poor Privacy | | | | | **HPO** - Good Utility Poor Privacy | | | |
| Main Task Pred. | Female ✓ | Male ✓ | Male ✓ | | Main Task Pred. | 30-39 ✓ | 3-9 ✓ | 40-49 ✓ |
| Priv. Task Pred. | White ✓ | East-Asian ✓ | Black ✓ | | Priv. Task Pred. | Female ✓ | Female ✓ | Female ✓ |
| **Noisy** - Poor Utility Good Privacy | | | | | **Noisy** - Poor Utility Good Privacy | | | |
| Main Task Pred. | Male ✗ | Female ✗ | Male ✓ | | Main Task Pred. | 3-9 ✗ | 20-29 ✗ | 40-49 ✓ |
| Priv. Task Pred. | Black ✗ | M. Eastern ✗ | Latino ✗ | | Priv. Task Pred. | Male ✗ | Male ✗ | Male ✗ |
| **Multi-Obj HPO (Ours)** Good Utility Good Privacy | | | | | **Multi-Obj HPO (Ours)** Good Utility Good Privacy | | | |
| Main Task Pred. | Female ✓ | Male ✓ | Male ✓ | | Main Task Pred. | 30-39 ✓ | 3-9 ✓ | 40-49 ✓ |
| Priv. Task Pred. | Black ✗ | M. Eastern ✗ | Latino ✗ | | Priv. Task Pred. | Male ✗ | Male ✗ | Male ✗ |

Figure 7: Example of the main and private task inference from models trained using classic HPO, the noise injection baseline, and our multi-objective HPO method.

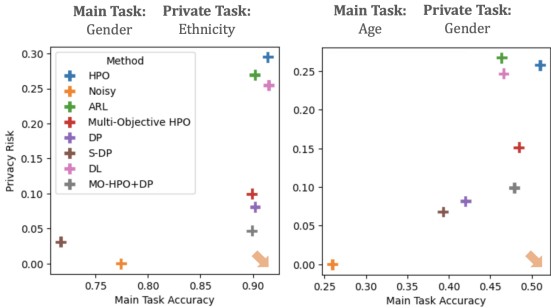

Figure 8: Privacy-utility trade-off of our method (Multi-Objective HPO ✚) and the combination of our method and the differential privacy mechanism (MO-HPO+DP ✚) and the baselines for two combinations of main and private task on the VGG-16 Architecture. The arrow represents the best possible privacy-utility trade-off: High main task accuracy and low privacy risk.

We then illustrate the privacy-utility trade-off of our method compared to the baselines in Fig. 8, for two combinations of main and private tasks. Ideally, we would like to reach both high main task accuracy and low privacy risk, i.e., be as close as possible to the lower right corner of the plot, as indicated by the arrow. The models trained using our multi-HPO method, depicted in red, offer a very good privacy-utility trade-off, in a similar range as the DP baseline, depicted in purple, whereas the combination of our method and DP clearly yields the best privacy-utility trade-off.

We then report results of experiments conducted in the same manner and against the same 6 baselines, but using additional architectures and datasets. We first evaluate our method using the same dataset, FairFace but using a different architecture for the embedding network: Inception-ResNet (Szegedy et al., 2016), with a latent space of size 512. We report the results for 6 combinations of main and private tasks in Table 4.

We then evaluate our method on the VGG16 architecture but this time using the CelebA dataset (Liu et al., 2015), which contains 202,599 face images with 40 labeled soft-biometrics. Because of the varying label quality of the CelebA dataset, we select 3 categories that have been shown to have high consistency across annotators (Wu et al., 2023): Gender, Gray Hair, Glasses, wich we use as options for main and private tasks. All three categories are binary classes. We report the results for 4 combinations of main and private tasks in Table 5.

## A.3 Alternate Plots

Table 4: Privacy-Utility trade-off comparison with four baselines on the FairFace dataset and Inception Res-Net architecture. The higher the main task accuracy (MA), the better, and the lower the privacy risk (PR), the better.

| | | HPO | | Noisy | | DL | | PCA | | MO-HPO (Ours) | | MO-HPO+PCA | |
|---|---|---|---|---|---|---|---|---|---|---|---|---|---|
| M. Task | P. Task | MA ↑ | PR ↓ | MA ↑ | PR ↓ | MA ↑ | PR ↓ | MA ↑ | PR ↓ | MA ↑ | PR ↓ | MA ↑ | PR ↓ |
| Gender | Ethnicity | 0.915 | 0.036 | 0.814 | 0.000 | **0.918** | 0.09 | 0.888 | **0.002** | 0.908 | 0.029 | 0.912 | 0.025 |
| Gender | Age | 0.914 | 0.112 | 0.809 | 0.001 | **0.918** | 0.143 | 0.887 | **0.061** | 0.887 | 0.100 | 0.877 | 0.090 |
| Ethnicity | Gender | **0.651** | 0.080 | 0.371 | 0.000 | 0.646 | 0.161 | 0.535 | 0.026 | 0.623 | 0.083 | 0.588 | 0.061 |
| Ethnicity | Age | **0.649** | 0.085 | 0.368 | 0.000 | 0.500 | 0.071 | 0.535 | 0.014 | 0.613 | 0.014 | 0.636 | **0.007** |
| Age | Gender | **0.549** | 0.216 | 0.219 | 0.036 | 0.509 | 0.236 | 0.348 | **0.068** | 0.513 | 0.150 | 0.517 | 0.109 |
| Age | Ethnicity | 0.545 | 0.144 | 0.220 | 0.021 | **0.547** | 0.166 | 0.343 | **0.027** | 0.527 | 0.103 | 0.518 | 0.049 |

Table 5: Privacy-Utility trade-off comparison with four baselines on the CelebA dataset on the VGG16 architecture. The higher the main task accuracy (MA), the better, and the lower the privacy risk (PR), the better.

| | | HPO | | Noisy | | DL | | PCA | | MO-HPO (Ours) | | MO-HPO+PCA | |
|---|---|---|---|---|---|---|---|---|---|---|---|---|---|
| M. Task | P. Task | MA ↑ | PR ↓ | MA ↑ | PR ↓ | MA ↑ | PR ↓ | MA ↑ | PR ↓ | MA ↑ | PR ↓ | MA ↑ | PR ↓ |
| Gender | Eyeglasses | **0.989** | 0.367 | 0.822 | 0.000 | 0.985 | **0.218** | 0.980 | 0.223 | 0.986 | 0.228 | 0.985 | **0.218** |
| Gender | Gray Hair | **0.989** | 0.353 | 0.825 | 0.000 | 0.986 | 0.189 | 0.980 | 0.189 | 0.986 | **0.187** | 0.985 | 0.189 |
| Eyeglasses | Gender | **0.996** | 0.225 | 0.801 | 0.000 | 0.992 | 0.188 | 0.935 | **0.064** | 0.991 | 0.224 | 0.991 | 0.188 |
| Gray Hair | Gender | **0.959** | 0.393 | 0.500 | 0.180 | 0.951 | 0.002 | 0.855 | 0.242 | 0.951 | **0.001** | 0.951 | **0.001** |

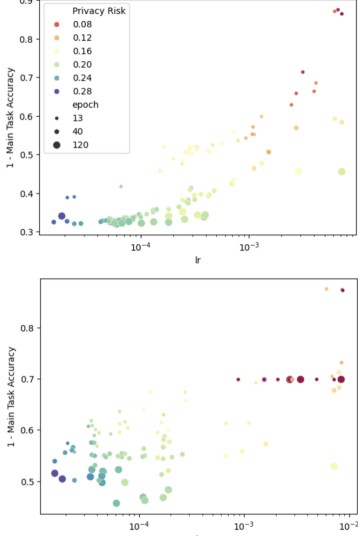

Fig. 9 shows an alternate version of Fig. 4 but we instead plots the main task accuracy as a function of the learning rate for all models trained using the Adam optimizer during two runs of multi-objective HPO for two different combinations of main and private tasks (Age-Gender at the top and Ethnicity-Age at the bottom). The size of the points indicates for how many epochs the model was trained. In this plot, however, the color depicts the privacy risk (the closer to red, the better). Once again, these plots highlight a strong negative correlation between the learning rate and the privacy risk.

Figure 9: Privacy risk as a function of the learning rate for all models trained using the Adam optimizer during two runs of multi-objective HPO for two different combinations of main and private tasks: Age-Gender (top), and Ethnicity-Age (bottom).

