# OpenReview forum: "On the Impact of Hyper-Parameters on the Inference Time Privacy of Deep Neural Networks"
_TMLR — Rejected by TMLR_

### Review · Reviewer_3Vpf · 2025-12-17

**Summary Of Contributions:**

The paper studies the effect of hyperparameters on the privacy of neural network embeddings (features).
It focuses on information leakage from the embeddings at inference time.
The authors derive a novel "privacy risk" metric that quantifies unintended feature leakage in the worst case.
They then use multi-objective hyperparameter optimization (MO-HPO) to optimize the hyperparameters with respect to both the main task (accuracy) and privacy risk.
Finally, the paper analyzes the hyperparameter configurations found by MO-HPO and shows that the learning rate and weight decay have the largest effect on privacy risk.

Strengths:
- The idea of hyperparameters influencing the privacy of a deep learning model is novel.
- Definition of privacy risk is well motivated and interesting.
- Figure 3 strongly demonstrates that for similar accuracy the privacy risk varies only depending on the hyperparameters.
- The MO-HPO seemingly matches the performance of other privacy-preserving techniques only by tuning the hyperparameters.
- The method is compared against multiple reasonable baselines.

Weaknesses:
- The differentially private (DP) baseline does not satisfy DP; instead, it just injects Laplace noise into the embeddings.
- The DP and MO-HPO+DP baselines inject different amounts of noise rendering them incomparable.
- The tables (and relevant figures) are missing confidence intervals.
- The paper is at times hard to follow; the presentation could be improved.

**Additional Comments:**

1. The batch size range could be extended, the current range of 30--100 is very narrow. The upper limit should preferably be the dataset size to better understand the effect of batch sizes.
2. Typos:
    - Section 4.2: "distribtuion"
    - Figure 8 caption: "thr"

**Audience:**

Yes

**Audience Explanation:**

The paper demonstrates hyperparameters that result in similar accuracy can have a much larger effect on the privacy risk.
This is an important and underexplored direction that is likely of interest to researchers and practitioners of privacy-preserving machine learning.

**Broader Impact Concerns:**

None.

**Claims And Evidence:**

No

**Claims Explanation:**

The biggest concern is the claims about achieving similar privacy to differentially private methods.
Namely, the DP baselines do not satisfy DP.
Instead the claimed DP baselines just inject Laplace noise to the embeddings with a standard deviation of $\frac{1}{\epsilon}$ without discussion of how sensitivity is calculated.
Furthermore, the cited paper (Osia et al., 2020) even states: "... or differential privacy [18], they cannot be directly applied to our problem .."
Also, the DP and MO-HPO+DP baselines are incomparable: the former uses $\varepsilon = 0.5$ and the latter $\varepsilon = 1.2$.
These faults seem to render the DP comparison unsupported and misleading.

**Requested Changes:**

1. The DP baselines need to be fixed or claims about the method being comparable to state-of-the-art DP methods removed.
2. The comparability issue of DP and MO-HPO+DP baselines need to be fixed: they need to be compared with the same amount of noise injected to the embeddings.
3. Preferably at least the main results should be reported with confidence intervals.
4. The marker colors in Figure 4 can be difficult to disambiguate. Could you please add a version of these plots with main-task accuracy on the y-axis and privacy risk encoded by color?

---

### Review · Reviewer_Bu99 · 2025-12-18

**Summary Of Contributions:**

The paper looks at the effect of hyperparameter optimization (HPO) on inference time privacy for feature extractors (ie, omitting the classifier head and looking only at the embeddings) in image classification, where privacy is measured by the adversary's success in predicting sensitive feature values from the embeddings. The main idea is to include privacy as an additional objective besides accuracy for HPO, turning HPO into a multiobjective optimization problem (MO-HPO). The main claims are that MO-HPO enables reaching privacy-utility tradeoff competitive with the current SOTA with very little overhead, and that there exists hyperparameter combinations having significant impact on privacy, and in particular settings that are better for privacy with negligible utility cost. The paper includes empirical experiments on VGG16 on FairFace data and Inception-ResNet on CelebA data.

**Additional Comments:**

## Minor tweaks and additional comments (no comments needed about addressing these):

* Fig 3: there seems to be a point on the lower left side that should be included on the Pareto front?
* I do not understand why include, eg, discussion about input, output and training algorithm perturbation in the Related work in Sec 2.1 as these are all training data privacy papers, and none seem relevant for inference time privacy (and as there do exist inference time DP/noise addition papers that would make sense to discuss here, see the list above). I would focus the discussion clearly on the inference time privacy methods.
* Please update all the references to published versions, remove doubles, fix spellings etc.
* Table 4 is missing bolding for best privacy results.

**Audience:**

Yes

**Audience Explanation:**

The theme of the paper is interesting; inference time privacy is considerably less explored compared, eg, to training time privacy, and I am sure a systematic study on the effects of HPO on inference time privacy is valuable.

**Broader Impact Concerns:**

No broader concerns

**Claims And Evidence:**

No

**Claims Explanation:**

There are several insufficiently supported claims and statements in the current version, as well as details requiring clarifications. Most importantly:
* The main claim that only doing multiobjective HPO results in privacy essentially on par with SOTA seems somewhat dubious, as the results do not seem very consistent (see eg Table 2, Fig 8). This is issue is made worse by the lack of any measure of variability in results.
* It is unclear to me from the current version if the adversary is strong enough for making the main claims; this is essential, as the privacy guarantees from the proposed method are purely empirical, unlike in some of the existing methods (eg, inference time differential privacy, DP).
* The comparisons with existing methods are somewhat hard to parse, as all the experiments report only a single point per method on the privacy-utility tradeoff curve, and the points generally do not systematically match on one component (ie, what is missing is something like compare acc with fixed privacy level).
* There seem to be some existing work omitted from the discussion.
* There are some missing details on the experiments needed to evaluate the results.
* The claim that the proposed method is very light weight compared to existing methods, as it only requires MO-HPO is somewhat misleading: to optimize privacy in MO-HPO, one needs to empirically measure the privacy of the hypers, which in this case then requires a suitable adversary, eg, a neural network trained for reconstructing the sensitive features for each hyper configuration. While the issue is discussed up to a point in Sec 4.1.2., it is not clear to me from the current paper how much compute would be needed to train a powerful-enough adversary. Claiming that this is more light weight than, eg, adding some Gaussian noise to the embeddings does not seem justifiable.

**Requested Changes:**

Please fix/clarify the following:
* It would be good to mention that you focus on *inference time* privacy already in the paper title
* Include at least the following papers to the discussion and as baselines/attacks, or justify why they are not relevant:
	* Abuadbba et al. 2020: Can We Use Split Learning on 1D CNN Models for Privacy Preserving Training?
	* He et al. 2019: Model inversion attacks against collaborative inference.
	* He et al. 2021: Attacking and protecting data privacy in edge–cloud collaborative inference systems.
	* Mireshghallah et al. 2020: Shredder: Learning Noise Distributions to Protect Inference Privacy.
	* Mireshghallah et al. 2021: Not All Features Are Equal: Discovering Essential Features for Preserving Prediction Privacy.
	* Singh et al. 2021: DISCO: Dynamic and Invariant Sensitive Channel Obfuscation for deep neural networks.
	* Yang et al. 2022: Measuring Data Reconstruction Defenses in Collaborative Inference Systems.
* From the already cited work, please shortly justify the choice of baselines.
* Sec 3.2, Eq (2): On measuring the privacy risk, do you assume the adversary has always a uniform prior, or why would privacy risk approaching zero always mean random predictions?
* Sec 4.1.2.: please formalize the attack model construction in pseudo-code to make this clearer.
* Related to the previous 2 questions, do you assume the adversary has access to some data from a similar/different distribution as the inference time samples for training an attack model or how exactly this works?
* Sec 2.1: "these techniques suffer from drawbacks, such the fact that they can be detected (Rot et al., 2022) or even attacked (Osorio-Roig et al., 2021)...". Does this mean that you assume some kind of security by obscurity for the proposed method? To me it seems that if you want to show that MO-HPO does provide some privacy, you should use the strongest possible adversary tuned to attack the specific model found by MO-HPO with full knowledge of your method. Do you agree with this, and does this match your empirical setup?
* Related to the previous point, Sec 4.1.2: on approximating the optimal adversary: i) is there some known optimal adversary, and ii) can you quantify how good the approximation you use actually is?
* On the random Gaussian noise baseline: given that there is a privacy-utility tradeoff, I do not really see the point of adding so much noise that privacy breach reaches 0. While this gives some baseline for sure, it would seem more relevant to tune the noise to match whatever privacy level the proposed method reaches.
* On the DP & siamese fine-tuning baseline: related to the previous comment, it would also make more sense to match the privacy level to enable apples to apples comparisons on accuracy.
* On the comparisons more generally: instead of looking at single point in the privacy-utility space, it would be good to try and map the privacy-utility curve (eg, for noise-based methods sweep at least a couple of noise scales). As the results currently stand (see eg Table 2, Fig 8), it is often not possible to tell if a method is better than another one as the results differ in both accuracy and privacy.
* On the empirical results more generally: if not done yet, please run at least 3 seeds and report, eg, mean + standard error/min+max for all experiments.
* Sec 1: "Additionally, all mitigation methods come with an inherent privacy-utility trade-off, where better privacy comes at the cost of main task accuracy, and which is very difficult to tune in practice." & Sec 2.1: "these techniques suffer from drawbacks, such as \[...\] trade-off between main task accuracy and privacy, as most of these methods degrade utility significantly." This sounds like you claim that i) the MO-HPO would be free of the tradeoff, which seems nonsensical (although you could of course claim, eg, that it Pareto dominates the existing methods), and/or ii) that tuning the tradeoff with MO-HPO would be significantly easier than with existing methods. Do you actually claim that MO-HPO has these properties?
* When will the code be available?

---

### Review · Reviewer_BkrC · 2026-01-02

**Summary Of Contributions:**

### Contributions:

This paper investigates **unintended feature leakage** in deep neural network embeddings, focusing on whether embeddings trained for a biometric task (e.g., gender classification) leak sensitive attributes (e.g., ethnicity or age) at inference time.  The paper studies **training hyper-parameters** as a simple and general lever for improving privacy.

The main contributions are:

1. **First systematic study of hyper-parameters and privacy leakage**
   The paper provides the first large-scale empirical analysis of how standard training hyper-parameters affect unintended feature leakage in embeddings, measured via a strong adversarial inference setup .

2. **Multi-objective HPO for privacy–utility trade-offs**
   The authors formulate privacy-aware training as a **multi-objective, multi-fidelity hyper-parameter optimization** problem that jointly maximizes main-task accuracy and minimizes private-attribute inference, enabling explicit exploration of the privacy–utility trade-off without changing model architectures .

3. **Identification of privacy-critical hyper-parameters**
   The empirical analysis shows that **learning rate and weight decay** have the strongest and most consistent impact on privacy leakage, and that privacy-favorable configurations often generalize across different private attributes .


### Strengths

* Highly practical and easy to deploy (no architectural or training modifications).
* Clear, actionable insights into which hyper-parameters matter for privacy.
* Results generalize across tasks, datasets, and architectures.

### Weaknesses

* Experiments are limited to vision and biometric attributes.
* Focuses only on inference-time leakage, not training-data privacy (e.g., membership inference).
* Privacy guarantees are empirical rather than formal.
* Full HPO procedure may still be computationally expensive.

**Audience:**

Yes

**Audience Explanation:**

At least a subset of TMLR’s audience—particularly researchers interested in privacy in machine learning, representation learning, and empirical properties of deep learning systems—would find the findings of this paper of interest. The paper offers a practically motivated perspective on privacy risks in embeddings and provides empirical insights into how standard training hyper-parameters influence unintended information leakage, which is relevant to both academic research and applied ML practice.

**Broader Impact Concerns:**

The work is largely ethically positive, as it aims to reduce unintended privacy leakage in learned representations using practical training choices.
However, a few concerns merit brief acknowledgment:
1. The results are empirical and adversary-dependent, which may lead to a false sense of strong privacy guarantees if hyper-parameter tuning is interpreted as sufficient protection.

2. The focus on biometric and face-based attributes (e.g., gender, ethnicity) raises concerns about downstream misuse in surveillance or profiling, even when leakage is reduced.

3. The absence of formal privacy guarantees (e.g., differential privacy) limits suitability for high-stakes deployments.

These issues suggest the Broader Impact discussion should  state the limitations, appropriate use cases, and non-guarantees of the approach.

**Claims And Evidence:**

No

**Claims Explanation:**

While the empirical results are extensive, several claims—particularly regarding the generality of privacy-preserving hyper-parameters and their effectiveness beyond the evaluated settings—are stronger than what the evidence strictly supports. Privacy is assessed using learned adversaries rather than formal or worst-case guarantees, and experiments are limited to biometric vision tasks and a small set of architectures and datasets. As a result, the evidence supports the empirical observations within the tested scope, but does not fully justify the broader privacy claims made in the paper

**Requested Changes:**

### **Critical changes**

1.
   The paper should explicitly clarify—both in the introduction and conclusion—that the reported privacy improvements are **empirical, adversary-dependent, and restricted to the evaluated datasets, architectures, and threat models**. Several statements currently risk being interpreted as providing broader privacy guarantees.
Please revise the wording of the main claims to explicitly state the scope and limitations, and add a short paragraph discussing what is *not* guaranteed by the proposed approach.

2.
   Table 2 shows that **DP-based baselines often achieve lower privacy leakage than MO-HPO**, and in some cases comparable or better utility. While this is briefly acknowledged, it is not sufficiently reflected in the overall framing of the contribution.
  Please:

   * Explicitly discuss Table 2 in the main text and clarify that MO-HPO does **not** outperform DP in general.
   * Reposition MO-HPO as **complementary to DP** (as also evidenced by the MO-HPO+DP results), rather than as a competing or replacement approach.
   * Clearly state in which scenarios MO-HPO alone is expected to be preferable (e.g., when DP cannot be applied or formal guarantees are not required).

3.
   The paper claims that certain hyper-parameters (notably learning rate and weight decay) “generally preserve privacy.” However, evidence is limited to **face-based biometric tasks, two datasets, and a small number of architectures**.
   Maybe (a) explicitly limit these claims to the studied settings, or (b) provide additional experimental evidence supporting broader generality (e.g., another task or modality).

4.
   The paper shows strong correlations between learning rate, weight decay, and privacy leakage, but it is unclear **what drives this effect**. In particular, it is not clear whether reduced leakage is:

   * merely a side effect of reduced main-task performance,
   * due to stronger regularization and smoother representations,
   * or related to convergence to different local minima.

    Please:
   * Add a dedicated discussion explicitly distinguishing correlation from causation.
   * Include additional analysis controlling for main-task accuracy (e.g., compare models with matched accuracy but different leakage).
   * If possible, add diagnostics (e.g., representation norms, margin distributions, or training dynamics) to support or refute proposed explanations.

5.
   The submission states that code will be released, but no code or link is currently provided.  Include a code repository (or at minimum detailed scripts, hyper-parameter configurations, and random seeds) to support reproducibility and transparency, in line with community best practices.



### **Changes that would strengthen the work**

6.
   The paper motivates MO-HPO as a practical alternative to complex privacy mechanisms, but it remains unclear **when this approach should be used in practice**.
Maybe add a short subsection explicitly answering:

   * In which deployment scenarios is MO-HPO preferable to DP or adversarial training?
   * What level of privacy risk reduction should practitioners realistically expect?

7.
   Privacy is evaluated using trained adversaries with specific capacities.
It would be ehlpful to add discussion (or a small experiment) showing how results might change under stronger adversaries (e.g., larger classifiers, additional auxiliary data), or clearly state why the chosen adversary is sufficient for the claims made.

8.
   All experiments focus on biometric face analysis.
Even a single additional experiment outside face biometrics (e.g., a non-biometric vision task or another modality) would substantially strengthen claims about the broader relevance of hyper-parameter–privacy interactions.

9. Please include a concise summary box or paragraph clarifying:

   * What privacy risks hyper-parameter tuning can mitigate,
   * What risks it does not address,
   * How it should be used alongside (not instead of) formal privacy methods.

---

### Decision · Action_Editor_RvtK · 2026-03-26

**Recommendation:** Reject

**Additional Comments:**

I recommend rejection of the current version but recommend allowing submitting a major revision.

Required revisions:

1. The authors should go through all claims made in the paper (and especially those in the Contributions section) and add cross-references to the evidence provided.
2. The authors should present their evaluations more clearly in terms of the privacy-utility tradeoff. Now these are often considered separately which makes the results hard to interpret.
3. The authors should consider reworking all their plots to use privacy-utility as the two axes and using different colours or symbols to highlight the objects of comparison (such as different hyperparameter values or alternative algorithms). This would make it obvious which choices are on the Pareto front and which are not.
4. The authors should consider any remaining reviewer comments (also see below).

Suggested revisions:

5. I would encourage the authors to consider making more use of the term "unintended feature leakage" instead of "privacy" to avoid confusion. At the very least, this should be repeated in results, tables, figure captions and claims. Many readers will not read the paper from the beginning to the end so mentioning this once is clearly not enough.


Additional comments from reviewers in their Official Recommendations:

Reviewer 3Vpf:

> The authors managed to address the main issues raised by me. Most importantly, they removed the claims about the method achieving results comparable to Differential Privacy (DP). As a minor note, Figure 8 still includes a DP method.
>
> However, I am not completely convinced about the causality of the claims. The results could also be interpreted as that well trained models have smaller privacy risks and these models are trained with smaller learning rates. Figure 9 (top) corroborates this. Figure 9 (bottom) looks more reasonable, but I am not completely convinced by this alone. The effect could have been more strongly demonstrated by showing the same data with confidence estimates, or by including more models and/or datasets, as requested by some of the reviewers.
>
> Furthermore, after removing the DP baselines and comparison, it is not anymore clear from Table 2 what the data says. This should have been clearly communicated in the main text and in the caption in the revised version.

Reviewer Bu99:

> While the authors have addressed several issues in the original manuscript, I still feel that there are several claims lacking support. The most important issue is that the proposed method (trying to optimize hyperparameters w.r.t. both accuracy/loss and privacy) is claimed to give performance on par with the actual privacy methods (highlighting that hyperparameter tuning is in some way more light weight than more complex methods, see the discussion on this in the comments), while the results are somewhat ambiguous. Most results are still lacking error bars, and the results with error bars (Table 2) are mixed, with the proposed approaches clearly failing in one column. In some cases it also seems hard to establish how different method perform relative to each other, as they clearly differ in both utility and privacy.

**Audience:**

Yes

**Audience Explanation:**

All reviewers consider that at least some individuals would be interested in the findings of the paper.

**Claims And Evidence:**

No

**Claims Explanation:**

Two out of three reviewers think that the paper is not supported by sufficient evidence.

I am especially concerned about the clarity of the evidence. The authors should go through all claims made in the paper (and especially those in the Contributions section) and add cross-references to the evidence provided. Furthermore, I would encourage the authors to rework all their plots to use privacy-utility as the two axes and using different colours or symbols to highlight the objects of comparison (such as different hyperparameter values or alternative algorithms). This would make it obvious which choices are on the Pareto front and which are not.

**Resubmission Of Major Revision:**

The authors may consider submitting a major revision at a later time.